# Combining Anomaly Detection and Supervised Learning for Medical Image Segmentation

**Julius C. Holzschuh**[1,2]                                JULIUS.HOLZSCHUH@DKFZ.DE
**David Zimmerer**[1]                                        DAVID.ZIMMERER@DKFZ.DE
**Constantin Ulrich**[1]                                    CONSTANTIN.ULRICH@DKFZ.DE
**Michael Baumgartner**[1]                                  MICHAEL.BAUMGARTNER@DKFZ.DE
**Gregor Koehler**[1]                                       GREGOR.KOEHLER@DKFZ.DE
**Rainer Stiefelhagen**[2]                                  RAINER.STIEFELHAGEN@KIT.EDU
**Klaus Maier-Hein**[1]                                     K.MAIER-HEIN@DKFZ.DE

[1] *Medical Image Computing, German Cancer Research Center (DKFZ), Heidelberg, Germany*
[2] *Karlsruhe Institute of Technology, Karlsruhe, Germany*

**Editors:** Accepted for publication at MIDL 2023

## Abstract

Fully-supervised machine learning has been established as an effective method for medical image segmentation. However, it requires large amounts of expert-annotated data, which can be a bottleneck in certain applications. Unsupervised methods like anomaly localization have proven their potential without relying on any labeled data, making them potentially much more scalable than fully supervised methods. Despite their scalability advantages, unsupervised and self-supervised methods have not yet fully reached the performance level of fully supervised models. As a first step to close this gap, we propose an approach that combines both concepts. We fine-tune a pre-trained anomaly localization model, namely a self-supervised denoising auto-encoder, using varying amounts of labeled training data in a supervised manner. Overall this approach exhibits superior performance compared to a model trained from scratch, especially in a low labeled training data regime.

**Keywords:** semi-supervised segmentation, self-supervised learning, anomaly localization

## 1. Introduction

Supervised machine learning methods, such as nnUNet (Isensee et al., 2021) have been wildly successful for medical image segmentation tasks. However, obtaining expert-labeled training data can be both difficult and expensive, limiting the scalability of such models. Pre-training has been proven to be a promising technique to overcome this challenge, leveraging large amounts of unlabeled data to learn generic features and consecutive fine-tuning on specific tasks with much less labeled data. However, pre-training is still not that frequently used for medical imaging, as the tasks involved are often very specific and potentially usable data is mostly segmented across multiple smaller datasets. Prevailing practices often limit the training of models and supervised pre-training to just one or a few similar datasets, which fails to take advantage of the potential synergy from other annotated data that could be available across different datasets. Although numerous self-supervised methods have emerged in the past, they often remain designed for tasks that are very different from the downstream tasks, making them not directly applicable to medical image segmentation.

Here, anomaly localization is more closely related to the downstream segmentation task compared to other self-supervised methods. However, despite its promising zero-shot anomaly segmentation performance, it was not yet considered as a pre-training approach. Thus, we here try to assess the potential of anomaly localization in the context of pre-training medical image segmentation methods.

Hence, we propose a novel approach that combines unsupervised anomaly localization with supervised fine-tuning for medical image segmentation, providing a promising solution to the challenge of obtaining large amounts of expert-labeled data. The first step of our study consists of training a UNet to perform denoising on healthy brain MRI images, using a well-established method for anomaly localization as the basis for our approach (Kascenas et al., 2023). In a subsequent step, we utilize this pre-trained model and assess the segmentation performance using varying amounts of labeled training data. We initially evaluated its performance in the binary tumor segmentation scenario to allow a fair comparison to anomaly localization methods, before demonstrating its transferability to a multi-class downstream task.

## 2. Materials and Methods

**Experimental setup** Experiments were conducted using the publicly available BraTS 2021 dataset (Baid et al., 2021), which comprises 1251 patient samples. Each sample includes a T1, T1Gd, T2 and T2-FLAIR sequence. The dataset was partitioned into three sets for training (n=938), validation (n=62), and testing (n=251). For comparability reasons, splits were kept the same as in (Kascenas et al., 2023). Prior to training, a slice-wise downsampling was performed to a resolution of 128x128, and the 99th percentile foreground voxel intensity was used to scale each individual modality of each scan. The anomaly localization model was trained solely on healthy slices. As architecture, a UNet with three stages for down and upsampling was used.

**Unsupervised Anomaly detection** In analogy to (Kascenas et al., 2023) the model was trained using a denoising task. Here, the best performing setup from (Kascenas et al., 2023) was chosen, i.e. Gaussian noise sampled at a low resolution of 16x16 on a per-pixel basis and then upsampled using bi-linear interpolation to the input resolution of 128x128 which was then added to the image. In order to prevent consistent upsampling patterns, the generated noise was randomly shifted. A more detailed description on the denoising implementation can be found in (Kascenas et al., 2023).

**Supervised finetuning** For the supervised task, models were trained using a combination of soft dice and cross entropy loss. While training models were selected and saved based on lowest validation loss. Saved models were then evaluated on the test dataset.

## 3. Experiments and Results

Results for a different number of labeled data samples (3D MRI Brain scans) as training data are shown in Figure 1. The lower green line and upper red line in the figure indicate the unsupervised and supervised baseline, respectively, as proposed by (Kascenas et al., 2023) on the same split. Displayed metrics for the binary case were collected using the

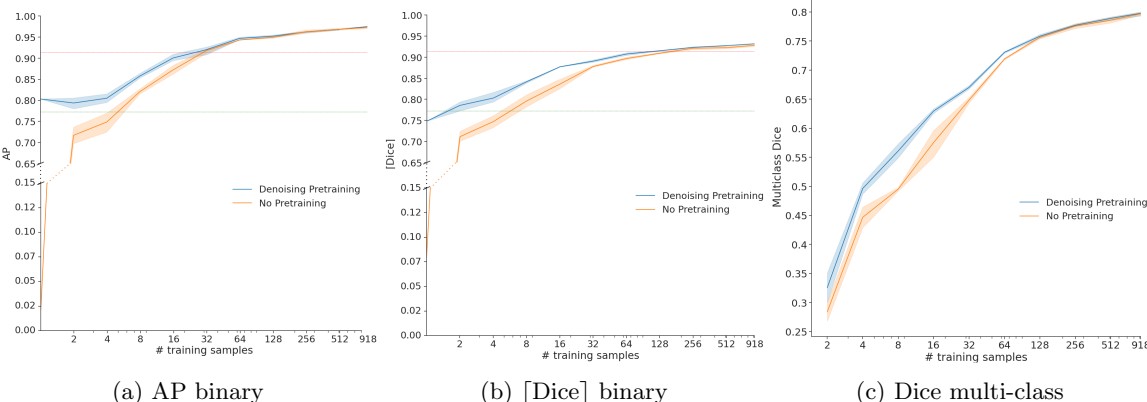

(a) AP binary      (b) ⌈Dice⌉ binary      (c) Dice multi-class

Figure 1: 1(a) and 1(b) present the average precision (AP) and ⌈Dice⌉ for binary segmentation, respectively. In step 0, the original head was utilized. Additionally, Figure 1(c) displays the mean Dice Score across all three tumour classes except the foreground. The horizontal axis of all graphs is presented on a logarithmic scale.

same implementation as proposed for the unsupervised baseline.

For multiclass segmentation, Dice Score was calculated patient wise for each 3D scan. Mean values were then calculated over all patients for each class individually. In Figure 1(c) the mean of all three classes (GD-enhancing tumor, peritumoral edema, non-enhancing tumor core) is presented while excluding the background class.

## 4. Discussion and Conclusion

In summary, the results indicate that using an anomaly localization task as pre-training can improve segmentation models, particularly for small amounts of training data. As expected the benefit of the pre-training diminishes as the number of training samples increases. However, the unsupervised baseline is easily outperformed with only a small number of labeled training samples. The not-pretrained baseline already exhibits a quite strong performance for only few training samples which can be partially attributed to the utilization of our nnU-Net-inspired (Isensee et al., 2021) data augmentation techniques, as well as to the relatively low complexity of the BraTS dataset (and our baseline outperforms the baseline presented in (Kascenas et al., 2023)). As this was primarily a proof-of-concept that anomaly localization can indeed be used as and be benefical as pretraining methods, we did not yet benchmark it to other pre-training schemes for medical image segmentation. Furthermore as anomaly localization is currently typically conducted slice-wise in 2D (Zimmerer et al., 2022), it is still unclear whether this denoising approach can be effectively extended to 3D settings and if the benefits extend beyond brain MRI or still hold in a more complicated setup like (Isensee et al., 2021) with extensive ensembling and postprocessing. However, here we have shown that self-supervised anomaly localization methods can be effectively used as pre-training, and present a promising comprise between fully-supervised and unsupervised methods, that might be especially beneficial for the medical imaging domain.

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
