# OpenReview forum: "Combining Anomaly Detection and Supervised Learning for Medical Image Segmentation"
_MIDL.io/2023/Short_Paper_Track — MIDL 2023 Short paper track Poster_

### Official Review · Reviewer_2GpQ · 2023-04-10
**Unsupervised pretraining can help quite a bit**

**Rating:** 7
**Confidence:** 5

**Review:**

In the natural image domain, it is well known that unsupervised pretraining can help quite a bit with classification and segmentation models (pretty much closing the gap with supervised pretraining). In medical imaging, there are fewer works. In this short paper, the authors use a denoising pretraining task to show fairly decent improvements in tumor segmentation using BraTs data, particularly in the low training data regime. The paper is well written and results include very informative, comprehensive curves showing performance vs N. I think this work could lead to good discussion at the conference.

One small point: how is denoising anomaly detection? I would change the title...

---

### Official Review · Reviewer_FJcw · 2023-04-25
**interesting idea and well-written paper**

**Rating:** 8
**Confidence:** 3

**Review:**

-	The approach of combining unsupervised anomaly localization with supervised fine tuning for medical image segmentation seems useful. The work is also original, to my knowledge.
-	The results indicate promising performance, demonstrating gains with small numbers of training examples.
-	Overall, the study appears well-conducted and includes a thoughtful discussion
-	As the authors acknowledge, comparison to other pre-training methods would be a useful future step